# Effects of Hot Extrusion Temperature on Mechanical and Corrosion Properties of Mg-Y-Zn-Zr Biological Magnesium Alloy Containing W Phase and I Phase

**DOI:** 10.3390/ma13051147

**Published:** 2020-03-05

**Authors:** Xiaofeng Wu, Chunxiang Xu, Jun Kuan, Zhengwei Zhang, Jingshan Zhang, Wenfu Yang

**Affiliations:** College of Materials Science and Engineering, Taiyuan University of Technology, Taiyuan 030024, China; Wuxiaofeng2012@126.com (X.W.); Kuan2012@126.com (J.K.); zhangzhengwei1225@126.com (Z.Z.); jinshansx@tom.com (J.Z.); Yang2014@126.com (W.Y.)

**Keywords:** magnesium alloy, hot extrusion, mechanical property, corrosion properties, dynamic recrystallization

## Abstract

The previous study conducted on the as-cast Mg-2Y-1Zn-0.6Zr alloy showed that the tensile strength, yield strength and elongation of the as-cast alloy were 245 MPa, 135 MPa and 14.4%, respectively. In order to further explore the potential of the material, the hot extrusion process of variable temperature (250 °C, 300 °C and 350 °C) was carried out on the basis of the as-cast alloy. After hot extrusion, the mechanical properties of the material have been greatly improved compared with as-cast alloy. The tensile strength, yield strength and elongation of the extruded alloy reached 327 MPa, 322 MPa and 24.9%, respectively. The reason for the significant improvement of material properties is mainly due to the dynamic recrystallization during thermal processing, which greatly fines the grains of as-cast alloy. Moreover, the experimental results shown that the corrosion performance of the alloy after hot extrusion at 300 °C is also optimal.

## 1. Introduction

Magnesium alloys have attracted increasingly more attention from many experts and scholars because of their high specific strength, specific stiffness, degradability, and good biocompatibility. The density of magnesium alloys is very small (1.74 g/cm^3^) compared to traditional biological materials such as titanium alloys and stainless steel, which gives it a higher specific strength and specific stiffness [1]. Given the fact that the modulus of elasticity of magnesium alloys (41–45 GPa) is the closest to that of human bones (28 GPa), it effectively avoids the stress shielding effect compared to stainless steel, titanium alloys, organic materials, etc., which grants it good biocompatibility [2]. However, magnesium alloys also have their own innate deficiencies, including low strength, poor plasticity, and fast corrosion rate. These problems have been plaguing many experts and scholars. Therefore, how to improve the toughness and corrosion resistance of magnesium alloys has become the focus of many experts and scholars at home and abroad. The Mg-Y-Zn-Zr series of alloys have attracted much attention from many scholars because of their superior mechanical properties and good corrosion resistance. The second phase species formed by Mg, Zn and Y element presenting in the Mg-Zn-Y alloy system are characterized as icosahedral quasicrystal I phase (Mg_3_Zn_6_Y_1_), cubic crystal phase W (Mg_3_Zn_3_Y_2_) and long-period stacking order structure LPSO phase (Mg_12_YZn) [3,4,5,6], whose presence or absence has a great influence on the mechanical properties of the alloy. In recent years, researches the Mg-Y-Zn-Zr series of alloys has been carried out by adding a high content of the Y element to introduce the long-period stacking ordered (LPSO) phase (Mg_12_YZn) to enhance the toughness of the alloy [7,8,9], but the higher-Y-content alloy can only be used as a structural material other than as a biomedical material. According to some research [10,11], the high content of the Y element (over 3 wt.%) can cause cytotoxicity to the human body. In addition, the as-cast microstructure of Mg-Y-Zn-Zr alloy is relatively coarse. Thus, proper thermal processing (hot extrusion) can not only greatly improve the mechanical properties of the alloy, but also improve the distribution of the second phase and improve the corrosion resistance of the alloy on the basis of the as-cast alloy. Especially, the impact of extrusion temperature on the dynamic recrystallization during hot extrusion is exceptionally significant. Therefore, this topic is dedicated to improving the toughness and studying the influences of extrusion temperature on the alloy´s corrosion resistance in hot extrusion.

## 2. Experimental Materials and Methods

The raw material used for this experiment was a high-purity magnesium ingot (99.5%), high-purity zinc (Zn) (99.99%), high-purity yttrium(Y) (99.99%), high-purity manganese (99.99%), and Mg-30%Zr master alloy. The melting was carried out in a high-frequency resistance furnace and protective gas (99% CO_2_, 1% SF_6_) was used to prevent the oxidation of magnesium melt during melting. After refining and holding for 20 min, the alloy melt was poured into a metal mold having a diameter of 20 mm. The sample needed to be homogenized before extrusion, so the polished magnesium ingot was placed in an electric resistance furnace at 350 °C for 12 h. The homogenization treatment of the sample allows the alloy elements in the magnesium ingot to be diffused uniformly. After the homogenization heat treatment, the magnesium ingot was placed in an abrasive tool with an extrusion ratio of 16:1, the extrusion rate was tentatively set at 60 mm/min, and the extrusion temperature was 250 °C, 300 °C, and 350 °C, respectively. Finally, an extruded rod with a diameter of 10 mm was obtained. The test of mechanical properties was mainly based on stretching. The standard specimen stretched by a DNS100 type electronic universal testing machine made by SINOTEST limited company in china Ji Lin province. The microscopic observation was carried out using an optical microscopy LeicaDM2500M made by Germany Tianjin Laike Optical Instrument Co., Ltd. The composition and morphology of the phases in the alloy were characterized by scanning electron microscopy (SEM, MIRA3-TESCAN) equipped with an energy spectrometer (EDS) and Y-2000 X-ray diffractometer (XRD) made by ShangHai China TESCAN company.

The simulated human body fluid used in the weightlessness experiment was SBF (simulated body fluid) solution, and the corrosion rate calculation formula is $\mathrm{CR}=8.76\times 10^{4}W/\mathrm{ATD}$, where A is the surface area of the sample, T is the test time of the sample, and D is the alloy density. The hydrogen evolution rate used in the hydrogen evolution experiments was $V_{H}=∆V/\left(A\cdot t\right)$, where ∆V is the volume of H_2_, A is the surface area of the sample, and t is the test time. Electrochemical experiments were carried out using the CS310 electrochemical workstation made by China Wuhan Coente Co., Ltd (Wuhan, China) of Wuhan Coente to test the impedance and time polarization curves of the extruded alloy. In order to better characterize the corrosion resistance of the alloy after extrusion, the corrosion performance of the extruded sample was characterized by roughness. The surface roughness test was tested by a VK-X200K laser confocal microscope made by Keyence Co., Ltd (Beijing, China).

## 3. Result and Discussion

### 3.1. Microstructure Analysis

Figure 1a,b shows optical microscope (OM) images of the as-cast Mg-2y-1-Zn-0.6Zr alloy and its corresponding grain size statistical chart. It can be seen from Figure 1a that the microstructure of the as-cast alloy is relatively coarse, whereas the second phase is distributed at the grain boundary or inside the crystal of the alloy. Some of the dense dotted second phase connecting and forming into strip distributed at the grain boundary. Subsequent XRD and EDS results show that the strip-shaped second phase at the grain boundary is mainly W phase, while the point-like second phase is quasicrystal I phase. The average grain size of the as-cast alloy is 77.88 μm (Figure 1b) and the size of most of the grains is concentrated between 50 and 90 μm. In order to refine grains, improve tissue defects and increase the microstructure density in the as-cast alloy. We firstly extruded the alloy at 250 °C. According to Figure 1c,d, we found that no significant dynamic recrystallization (DRX) occurred during extrusion. Only in a very small part of the region, did the grains begin to dynamically crystallization. This is because the formation and growth of the recrystallized nucleus requires atomic diffusion and high angle grain boundaries (or subgrain boundaries) moving to regions with high dislocation density, both of which require the necessary temperature to activate themselves to migrate, resulting to crystallization process can be completed absolutely [11]. When the extrusion temperature is 250 °C, the temperature is too low to stimulate the diffusion of atoms, so only partial dynamic recrystallization occurs in some high energy regions where the deformation is large and dislocation density is high. As the temperature decreases after extrusion, these local incompletely recrystallized grains are retained, which causes these grains are extremely fine. This indicates that the hot working temperature of Mg-2Y-1Zn-0.6Zr alloy is higher than 250 °C, so we continued to carry out another two hot extrusions at 300 °C and 350 °C. When the extrusion temperature is 300 °C, dynamic recrystallization occurs in most areas of the alloy, whereas some areas does not occur dynamic recrystallization (UnDRXed area), but to be elongated into a white strip along the extrusion direction. The average grain size of the extruded alloy is 1.62 μm and most of the crystal grains are concentrated in the range of 1 μm to 2.5 μm. As shown in Figure 1f, the microstructure of the alloy is greatly refined compared to 77.88 μm of the as-cast alloy, which significantly improves the strength and plasticity of the alloy.

When we continue to increase the extrusion temperature to 350 °C, most areas of microstructure also undergoes dynamic crystallization, while the area where dynamic recrystallization did not occur increased compared with that at 300 °C. Moreover, there are some abnormally coarse grains in the recrystallized area, so the average grain size of the alloy is 1.79 μm, which causes its average grain size is relatively higher than that (1.62 μm) of 300 °C. As shown in Figure 1h, the percentage of crystal grains having a grain size of between 2.5 μm and 3.5 μm in the as-extruded alloy at 350 °C is significantly higher than that at 300 °C. The reason for the abnormal growth of crystal grains is mainly due to the hindrance of which including inclusions, second phase particles (W and I phase) and the deformation texture to the grain growth process. They are distributed dipersely in the alloy (Figure 1d) when the extrusion temperature is low, while they will accumulate or dissolve in the Mg matrix at high temperature, which causes a small number of grains to get rid of the bondage of the inclusions or the second phase and get the priority to grow up. However, most of the grains blocked by the second phase or inclusions can’t grow, thus creating conditions for the growth of some abnormally coarse grains which reduce the strength, ductility and toughness of the alloy.

### 3.2. Phase Composition Analysis during Extrusion

The second phase in the Mg-2Y-1Zn-0.6Zr as-cast alloy is distributed in a dotted manner or short strip manner on the magnesium matrix. According to the results of EDS (Figure 3 and Table 1) and XRD (Figure 2a), these dispersed second phases mainly consisting of W phase and relatively less amount of quasi-crystal I phase. In high power electron microscopy at as-cast triangular boundary, it can be seen that there is no obvious boundary between the two second phases (Figure 2c), which indicates their growth relationship belongs to the symbiotic growth during crystallization process [12,13]. After hot extrusion, these dotted second phases become more continuous and distributed along the direction of extrusion on the α-Mg matrix (Figure 1d).

The W phase and quasi-crystal I phase belong to the hard and brittle phase according to some relevant researches [14,15]. The broken second phase (W phase and I phase) distributed on the magnesium matrix during the hot extrusion process effectively hinders the movement of dislocations and has a certain effect on the improvement of the material strength. Moreover, what more important is that these broken second phases hinder the migration of grain boundaries, thus hindering the abnormal growth of the grains caused by excessive temperature and helping to obtain fine grains during dynamic recrystallization. When the extrusion temperature is 250 °C, there is not enough energy to active dislocations and grain boundary movement due to the low extrusion temperature below the recrystallization temperature, so there is no much dynamic recrystallization occurs apparently. In addition, though the W phase and I phase are crushed into particles (Figure 3b), the effect of preventing dislocations and grain boundary movement is not effectively exerted. At 350 °C extrusion temperature, the grain boundary and dislocations get sufficiently high activation energy to get itself moved, but most of the crushed second phase has dissolved into the α-Mg matrix at relative higher extrusion temperature (Figure 3d), which results in the migration of grain boundary and dislocations not being hindered and contributing crystal grains grow into some coarse grains. When the extrusion temperature is 300 °C, the broken second phase is not dissolved into the crystal absolutely (Figure 3c), therefore effectively hindering the migration of the grain boundary and the growth of the crystal grains. We can see from Figure 3d, that’s why there is substantially no distribution of the broken second phase around the abnormally grown grains [14].

To further determine the composition of the second phase after extrusion, we performed a series of EDS dotting analyses (as shown in Figure 3 and Table 1). The W phase and I phase were detected in the extruded alloys at 250 °C and 350 °C, respectively. However, XRD did not detect the presence of the I phase at 300 °C, due to either a lower quantity of the I phase itself or the fact that the I phase was broken into fine particles after hot extrusion.

### 3.3. Extrusion Process Analysis

Hot extrusion is actually one of many metal thermal processing processes and essentially different from metal cold processing. The essential difference between hot working and cold working is temperature. A large amount of dislocation entanglement occurred inside the crystal grains after plastic deformation of metal during cold working, which resulted in work hardening [15]. However, as the deformation temperature was higher than the crystallization temperature in the hot extrusion process, the hardening process caused by plastic deformation and the softening process caused by dynamic recrystallization existed simultaneously.

Take the extrusion at 350 °C for example, as shown in the compression curve during hot extrusion (Figure 4), the stress increases with strain at first and the curve becomes gentle after reaching the maximum value σ_2_. This is because work hardening predominates before the peak σ_2_ that only partial dynamic recrystallization occurs in the metal, resulting the work hardening effect is greater than the softening effect of dynamic recrystallization. When the stress reaches the maximum value σ_2_, as the dynamic recrystallization accelerates, the softening effect is greater than the hardening effect and the stress begins to decrease. The work hardening effect caused by deformation tends to balance with the softening effect caused by dynamic recrystallization, therefore the work hardening rate is zero and the curve enters a stable state, finally the curve is close to a horizontal line that deformation can continue to occur under the action of stress σ_1_. Therefore, only partial dynamic recrystallization did occur during processing at 250 °C temperature, which cause the stress required for extrusion is relatively large. From the extrusion process, we can control the size of the subcrystal by adjusting the temperature that rapidly cooling in the initial stage of dynamic recrystallization and preserving the dynamically recrystallized structure to obtain a metal material with a very fine grain to increase the strength of the metal [16].

At higher temperatures (300 °C and 350 °C), although the alloy undergoes dynamic recrystallization in most areas while there are still a few areas (Figure 1e,g) where did’t occur dynamic recrystallization were extruded into strip. Moreover, the W phase and I phase are also extruded into a strip shape and distributed along the extrusion direction. The main reason for the absence of dynamic recrystallization in some areas is the extrusion stress inhomogeneity during deformation. As shown in the Figure 5, when the magnesium alloy bar is extruded, the surface metal flows slowly while the middle metal flows fast due to the frictional resistance of the die orifice. Moreover, the outer layer is pulled and the middle metal is pressed, which causes the magnesium alloy rod is unevenly stressed and the degree of deformation is not uniform during the extrusion process. Dynamic recrystallization can easily occur at some areas where the relative large deformation amount combined with a suitable deformation temperature. Those regions with large amount deformation combined with a suitable temperature above the hot working temperature of the material prioritize to occur dynamic recrystallization than other areas [8,17]. What’s more, because a large number of deformation dislocations used to form sub-crystals are also concentrated in a large deformation region, thus the driving force for crystallization in this area is larger. However, in some regions where deformation is not enough big to store the energy of recrystallization nucleation, which cause the dynamic recrystallization is not easy to occur (Figure 1e,g).

### 3.4. Mechanical Properties

On the whole, the mechanical properties of the alloy after extrusion have a significant improvement in strength and plasticity compared to the as-cast alloy, which is mainly due to the large grain refinement after extrusion. Fine grain strengthening is an extremely important strengthening method for metal materials. It can not only improve the strength of materials but also improve ductility and toughness, which is unmatched by other strengthening methods [18]. The σ_s_, σ_b_, and elongation of the as-cast Mg-2Y-1Zn-0.6Zr alloy were 135 MPa, 245 MPa, and 14.4%, respectively. The σ_s_, σ_b_, and elongation after extrusion could reach 322 MPa, 327 MPa, and 24.9%, respectively, but they were slightly different at different extrusion temperatures (Table 2).

At an extrusion temperature of 250 °C, the alloy had a high yield strength and tensile strength while the ductility was relatively low, even lower than that of the as-cast alloy. The main reasons for the poor plasticity of the alloy are as follows: (1) The dynamic recrystallization of the alloy was not fully performed, due to the low extrusion temperature; the work hardening during the plastic deformation process was not completely eliminated and a large number of high-density dislocations still existed in the crystal grains, resulting in the resistance to dislocation motion during deformation and an increase in deformation resistance, finally facilitating the strength of the metal while reducing the ductility and toughness. (2) After the metal was plastically deformed, a large internal stress generated due to lattice distortion caused the hardness and strength of the metal to increase while the ductility and corrosion resistance were lowered [19]. (3) The low deformation temperature led to the crushed second phase not dissolving into the matrix, so micro cracks easily induced at the second phase during the stretching process caused the material fracture and reduced the ductility of the material.

At 300 °C and 350 °C extrusion temperature, the strength of the alloy is lower than 250 °C while the ductility of the alloy has been greatly improved (Figure 6), which mainly attributes the relatively sufficient performance of dynamic recrystallization at higher temperatures. When the extrusion temperature is 300 °C, the average grain size of the alloy is 1.62 μm (Figure 1f). The fine grains make the strain difference between the inside of the grain and the grain boundary smaller and the deformation is more uniform, therefore it can withstand large amount of deformation before the material breaks, which contribute to a large elongation [20]. Moreover, cracks in the fine-grained metal are not easily generated and expanded and more energy is absorbed in the fracture process, thus the material exhibiting a higher toughness. When the extrusion temperature is 350 °C, the average grain size of the alloy is 1.79 μm. It is the relatively high extrusion temperature that lead to some abnormal coarse grain appears in the local area that result in the strength of the material reduced compared with that at 300 °C.

### 3.5. Tensile Fracture Analysis

In order to further explore the fracture mechanism of the material and fully understand the form of alloy failure, we analyzed the fracture morphology of the alloy under a high-power electron microscope. As shown in the schematic Figure 7, the dimple is actually the process of formation, expansion and connection of micropores during plastic deformation [21]. We can see from Figure 8a,b that the micro-fracture morphology of the as-cast and extruded alloys at different extrusion temperature. The fracture of as-cast alloy is basically plastic fracture that there are a large number of dimples on the fracture surface. The dislocations moved plastically along the slip surface under the action of large stress, and the stress was then concentrated when the dislocation moved to the vicinity of the second-phase particles, which caused the interface between the matrix and the second phase too pen to form micropores. Micropore expansion and connection were also the result of plastic deformation of the base metal. What caused the unstable fracture of the material was the metal between the adjacent micropores causing large plastic deformation when the micropores were expanded to a certain extent [22]. Therefore, we can see from Figure 8a that there were many small dimples distributed around some large dimples formed by the aggregation of small dimples, eventually leading to the fracture of the material.

The elongation of the extruded alloy was only about 10% at an extrusion temperature of 250 °C. As shown in Figure 9c,d, the form of failure of the alloy mainly manifested as brittle fracture where the fracture surface consisted of many tearing ridges and a small amount of dimples. The alloy exhibited good ductility (elongation is 25.4%) when the extrusion temperature was 300 °C, which is owed to the grains refined by dynamic recrystallization, and so does the size of the dimples compared to that of the as-cast alloy. Therefore, the fracture of the alloy at the extrusion temperature of 300 °C is a typical microporous aggregate fracture. Moreover, the second-phase particles are clearly visible at the bottom of the dimple and confirmed to be the W phase by EDS (Figure 9). When the extrusion temperature continued to rise to 350 °C, the ductility of the material was lowered due to the local coarse crystal grains because of the excessive temperature. According to the fracture theory of crack formation in the tensile test [23], the grain size has the following relationship with the crack propagation critical stress σ_c_, $\sigma_{c}\approx \frac{2\mu \gamma_{p}}{k_{y}}d^{-\frac{1}{2}}$, where *γ_p_* is the specific surface energy (the work consumed per unit area of surface area when the crack propagates), *K_y_* is the Petch slope, d is the grain size, and *μ* the is shear modulus. Any factors that can increase *σ_c_* can improve the plasticity of the material. Therefore, the alloy at the extrusion temperature of 300 °C shows higher strength and better toughness.

## 4. Effect of Extrusion Temperature on Corrosion Properties of Alloys

### 4.1. Weight Loss Experiment Results and Analysis

Figure 10 shows the weight loss corrosion rate of extruded Mg-2Y-1Zn-0.6Zr alloy after being immersed in SBF solution. The relationship at the corrosion rate of the alloy is E250 °C > E350 °C > E300 °C. The alloy extruded at 300 °C extrusion temperature has the best corrosion resistance 0.4601 < 0.5 mm/y and meets the requirements of bio-alloys. Based on existing researches [19,24], the undissolved second phase (W phase and I phase) on the matrix or the undynamic recrystallized area has a higher corrosion potential than that of the magnesium matrix, thus forming a primary cell that aggravates the corrosion rate of extruded alloy (at 250 °C extrusion temperature). At 300 °C extrusion temperature, the alloy undergoes relative sufficient dynamic recrystallization leads to the grains fine and uniform, which result to the corrosion performance of the alloy is the best. However, owning to the abnormal growth of crystal grains in the local region at 350 °C, the coarse microstructure reduces the corrosion resistance of the alloy.

### 4.2. Electrochemical Experiment Results and Analysis

In order to further explore the corrosion resistance of extruded alloys, Figure 11 shows the electrochemical polarization curves of the alloys after extrusion at different temperatures, whereas Table 3 shows the electrochemical parameters and the weight loss corrosion rate. The electrochemical polarization curve consists of an anodic polarization curve (left side) and a cathodic polarization curve (right side). The anode generates an oxidation reaction that dissolves the magnesium matrix: $\mathrm{Mg}\to {\mathrm{Mg}}^{2+}+2e^{-}$, while the cathode undergoes a hydrogen evolution reduction reaction: 2H_2_O + 2e^−^ → H_2_ + 2OH^−^ [25]. It can be seen from the polarization curve that as the extrusion temperature increases, the self-corrosion potential Ecorr moves first in the positive direction and then in the negative direction. The self-corrosion potential is the most positive when the extrusion temperature is 300 °C while the self-corrosion current Icorr of the alloy decreases first and then increases. Corrosion potential Ecorr reflects the degree of corrosion resistance of the alloy that the more positive the potential is, the more corrosion resistant the alloy becomes. Similarly, the self-corrosion current Icorr also reflects the corrosion rate of the alloy that the slower the corrosion rate of the alloy is, the better the corrosion resistance of the alloy get. When the extrusion temperature is 300 °C, the corrosion resistance of the alloy is best with the slowest corrosion rate, which is consistent with the static immersion test results.

The relationship between corrosion rate (*P_i_*) and corrosion current density (*I_corr_*) can be expressed by the following formula [25]:(1)$$P_{i}=22.85I_{corr}.$$

Based on the electrochemical parameters of the sample (*I_corr_*, *β_α_*, and *β_b_*), the polarization resistance (*R_P_*) is calculated from the following formula [26]:(2)$$R_{p}=\frac{\beta_{a}\beta_{b}}{2.3(\beta_{a}+\beta_{b})I_{corr}}.$$

In the polarization test, the alloy has the lowest corrosion rate at an extrusion temperature of 300 °C is 0.372 mm/y. The polarization resistance is inversely proportional to the corrosion rate. The larger the polarization resistance is, the smaller the corrosion rate become. Therefore, the alloy of the extrusion temperature 300 °C has the best corrosion resistance and the largest polarization resistance of 3.438 kΩ·cm^2^.

Figure 12c shows the electrochemical impedance spectroscopy and corresponding equivalent electrical circuit of the alloy at different extrusion temperatures. Electrochemical impedance spectroscopy contained a single high-frequency capacitive reactance loop caused by a double-layer charge transfer between the metal surface and the surface of the corrosive medium [25,26]. The high-frequency capacitance loop radius can reflect the corrosion resistance of the alloy at different extrusion temperatures. The larger the radius of the high-frequency capacitance loop are, the larger the resistance of charge transfer and the better the corrosion resistance of the material is [23,26]. The radius relationship of the high-frequency capacitive anti-arc was E300 °C > E350 °C > E250 °C. The results also conform to the conclusions of electrochemistry.

In addition, the Figure 12b show that extruded alloy at 300 °C extrusion temperature has the highest impedance value. The two peaks contained in frequency vs. degree (Figure 12a) corresponding two time constants, and the relationship of phase angle at medium frequency is E300 °C > E350 °C > E250 °C, which indicates the extruded alloy at 300 °C extrusion temperature has the best corrosion resistance again.

In order to understand the mechanism of the corrosion behavior of extruded alloy at different extrusion temperature further, the equivalent electrical circuit (EEC) for electrochemical impedance spectroscopy (EIS) plots was applied to fit the experimental data via ZSimpWin software. Figure 12c shows the EEC of different extruded alloys where Rs was the solution resistance between the reference electrode and the working electrode, Rct denoted the charge transfer resistance. Based on the proposed equivalent circuit models and the properties of the composite coating, the EIS curves were best fitted and the corresponding values of the equivalent circuit parameters are listed in Table 4. As we known, higher Rct values represent lower dissolution rate. The high Rct value (2198 Ω·cm2) of extruded alloy at 300 °C extrusion temperature suggests a low corrosion rate.

### 4.3. Immersion Corrosion Morphology Analysis

Figure 13 shows the surface morphology of the corroded alloy after removing corrosion products at different extrusion temperatures. We can conclude that the degree and manner of corrosion of the alloys at different extrusion temperatures were different.

The corrosion of the alloy at 250 °C was the most severe that large number of severe corrosion pits appeared on the surface of the substrate. This is mainly due to the following two reasons: (1) Since there are many regions where dynamic recrystallization did not occur, there is a potential difference between the dynamic recrystallized regions and undynamic recrystallized regions, thus generating many primary cells that cause galvanic corrosion of the material [27]. (2) The second phase (W phase and I phase) is not fully dissolved into the magnesium matrix due to the low extrusion temperature (Figure 1c), it is also a potential difference between them that causes galvanic corrosion. This can be seen from the previous polarization curve that the self-corrosion potential is low and the self-corrosion current is large, resulting to the faster corrosion rate.

When the extrusion temperature is 300 °C, there is little serious corrosion pits on the magnesium matrix. This is because the uniform equiaxed fine grain caused by relatively complete dynamic recrystallization impedes the spread of local corrosion. Some large corrosion pits appeared in the local area of the magnesium matrix at the extrusion temperature of 350 °C, but it was not as severe as the corrosion pit at 250 °C. This because the extrusion temperature is so high that some abnormal coarse grains appear locally in the alloy microstructure that cause corrosion to proceed more easily and large corrosion pits appear, thus reducing the corrosion resistance of the alloy.

### 4.4. ISCM Morphology Test and Analysis of Soaking Corrosion

Figure 14 is a three-dimensional topographical view of the extruded alloy at 250 °C extrusion temperature after corrosion immersion and corrosion product removal. In order to more accurately reflect the corrosion of the sample at different extrusion temperatures, we performed two different three-dimensional topographies at each extrusion temperature. As shown in the Figure 14a,b, the surface of the sample was severely corroded that a wide and deep corrosion pit was formed on the corrosion surface. The entire surface became uneven that its surface roughness Ra are 41μm (Figure 14a) and 34 μm (Figure 14b), the local maximum corrosion pit depth reaching 175 μm (Figure 14c) through scribing analyze, whereas the line roughness is 70 μm and 76 μm. The experimental results are basically consistent with the previous Immersion corrosion morphology.

Figure 15 is a three-dimensional topographical view of the de-corroded product of the extruded alloy at an extrusion temperature of 300 °C. The alloy mainly exhibited overall mild corrosion rather than severe localized corrosion. Therefore, it can be seen from its three-dimensional corrosion topography that the corrosion surface is relatively flat and the corrosion is relatively uniform due to the uniform fine grains, there is no more serious corrosion pit in the local area [28]. The line roughness curve also had no large undulations, both surface roughness Ra of Figure 15a,b were 9 μm while the line roughness was 8 μm.

Figure 16 is a three-dimensional topographical view of the de-corroded product of the as-extruded alloy at an extrusion temperature of 350 °C. It can be seen from Figure 16a that the alloy showed localized corrosion pits appearing locally, which is similar to that of the alloy at an extrusion temperature of 250 °C, but the depth of the corrosion pit in Figure 16a compared to that of Figure 14 was only 55 μm. Its surface roughness Ra = 23 μm, whereas the line roughness was 16 μm. Figure 16b shows a small undulation on the line roughness curve, which shows that the depth (6 μm) and width of the etch pit were relatively small, indicating that it is also typical local corrosion in which the surface roughness Ra = 14 μm and line roughness 9 μm.

The three-dimensional corrosion morphology at three extrusion temperatures and the corresponding line roughness and surface roughness also show that the corrosion was relatively uniform owing to the uniform fine grains when the extrusion temperature was 300 °C. Based on the above experimental data, we determined that the final extrusion temperature was 300 °C.

## 5. Conclusions

(1) After the as-cast alloy is hot extruded, the grain of the alloy is remarkably refined, mainly due to dynamic recrystallization during hot extrusion.

(2) During the extrusion process, the morphology and distribution of the second phase changed, but the species of the second phase did not change. Moreover, the second phase broken during the extrusion process acts to hinder grain boundary migration and refine grains.

(3) After extrusion at different extrusion temperatures, the optimum extrusion temperature of 300 °C is preferred, and the strength and elongation are greatly improved compared with the as-cast alloy that the yield strength, tensile strength and elongation are 322 MPa, 327 MPa and 24.9%, respectively. While the tensile strength increased by 35.1% and the yield strength increased by 72.9%.

(4) Through the corrosion test of the alloy after extrusion at different extrusion temperatures, several test results were synthesized that the corrosion resistance performance at different temperatures was: E300 °C > E350 °C > E250 °C.

## Figures and Tables

**Figure 1 materials-13-01147-f001:**
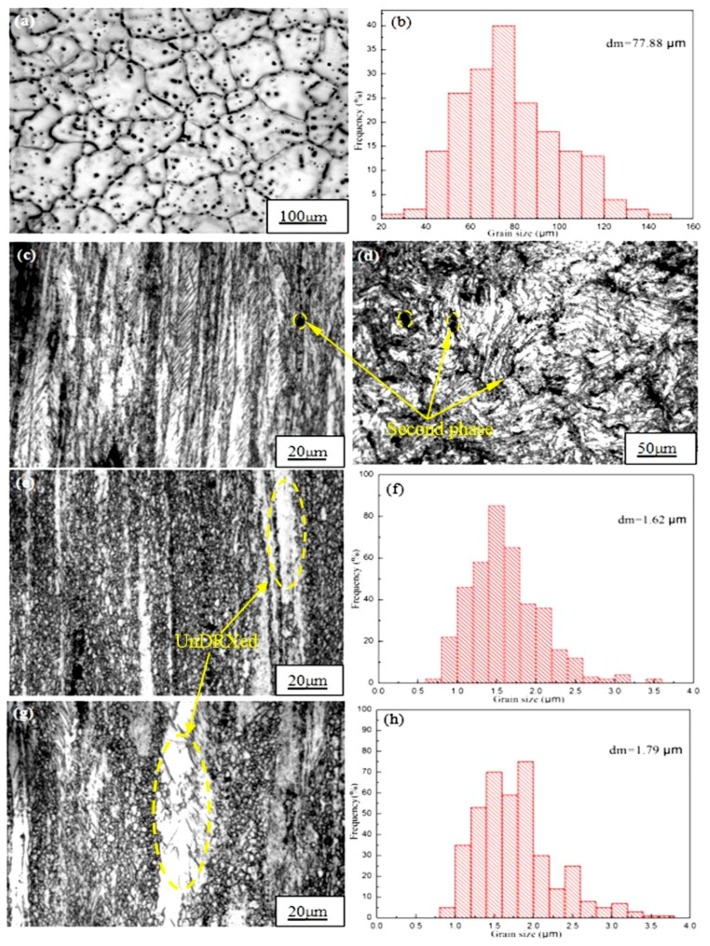
Optical micrographs and average grain size of alloy extruded at different temperatures: (**a**,**b**) As-cast; (**c**,**d**) 250 °C; (**e**,**f**) 300 °C; (**g**,**h**) 350 °C.

**Figure 2 materials-13-01147-f002:**
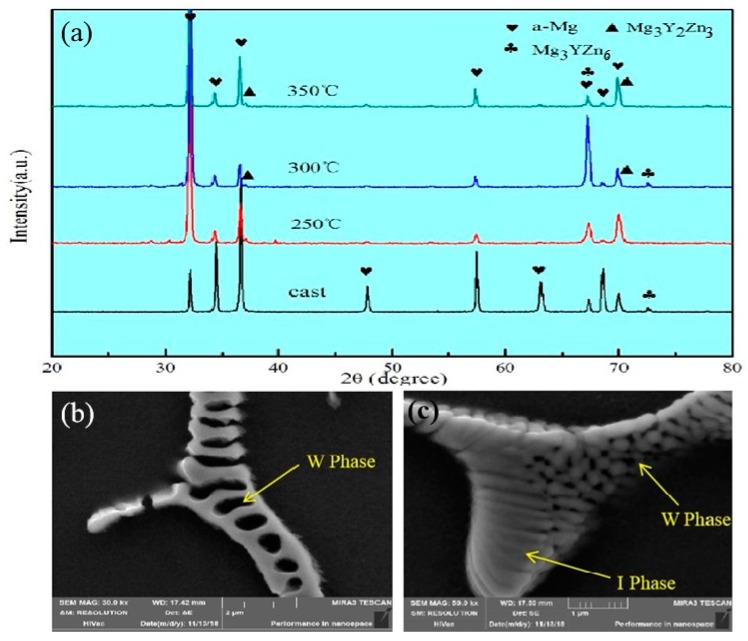
(**a**) XRD patterns of Mg-2Y-1Zn-0.6Zr alloy corresponding to the as-cast and as-extruded specimens; (**b**) micromorphology of W; (**c**)micromorphology of I.

**Figure 3 materials-13-01147-f003:**
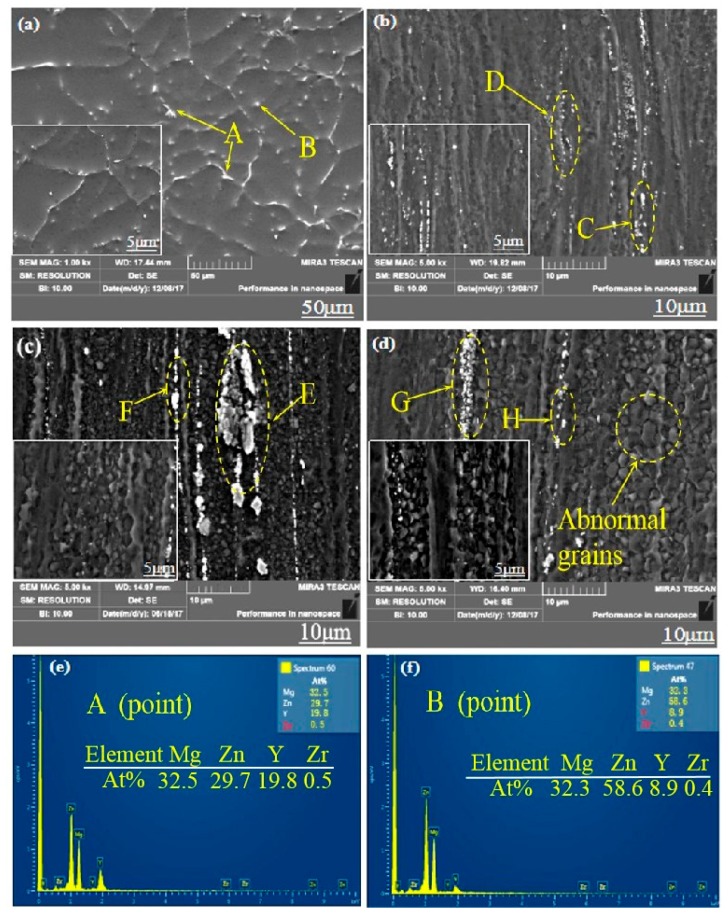
SEM micrographs of Mg-2Y-1Zn-0.6Zr as-cast and -extruded alloy: (**a**)As-cast; (**b**) 250 °C; (**c**) 300 °C; (**d**) 350 °C; (**e**,**f**) EDS spectra of extruded alloy.

**Figure 4 materials-13-01147-f004:**
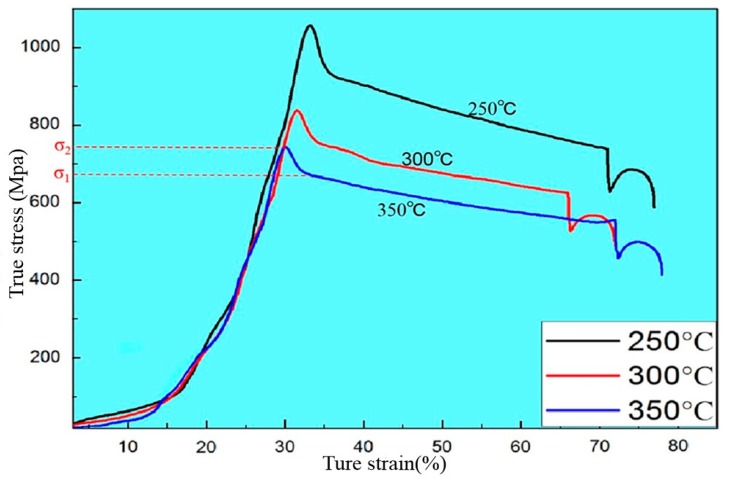
Typical compressive true stress-true strain curves of Mg-2Y-1Zn-0.6Zr alloy deformed at different temperatures.

**Figure 5 materials-13-01147-f005:**
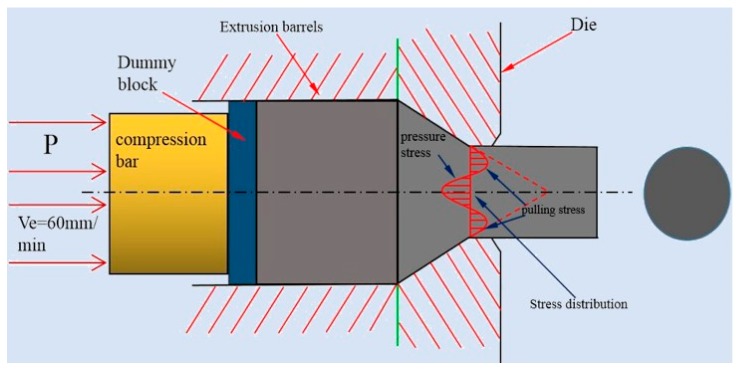
Schematic diagrams of extrusion.

**Figure 6 materials-13-01147-f006:**
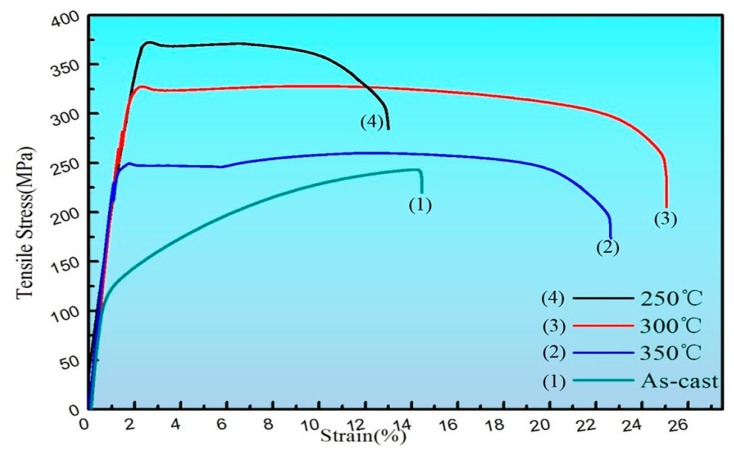
The tensile stress-strain curves of as-cast and as-extruded.

**Figure 7 materials-13-01147-f007:**
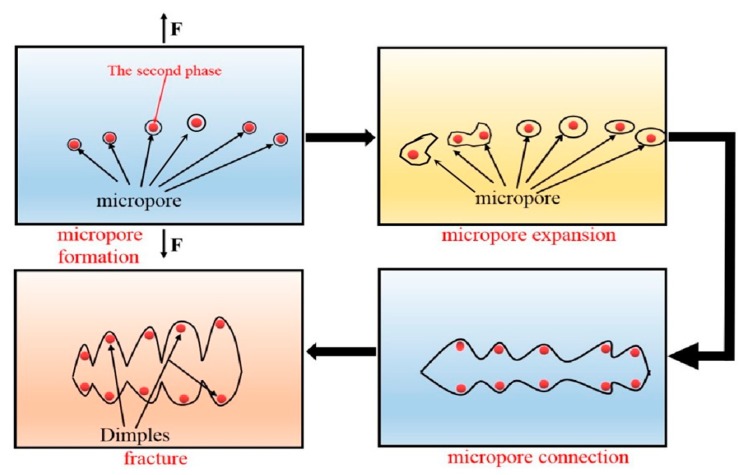
Schematic diagram of fracture.

**Figure 8 materials-13-01147-f008:**
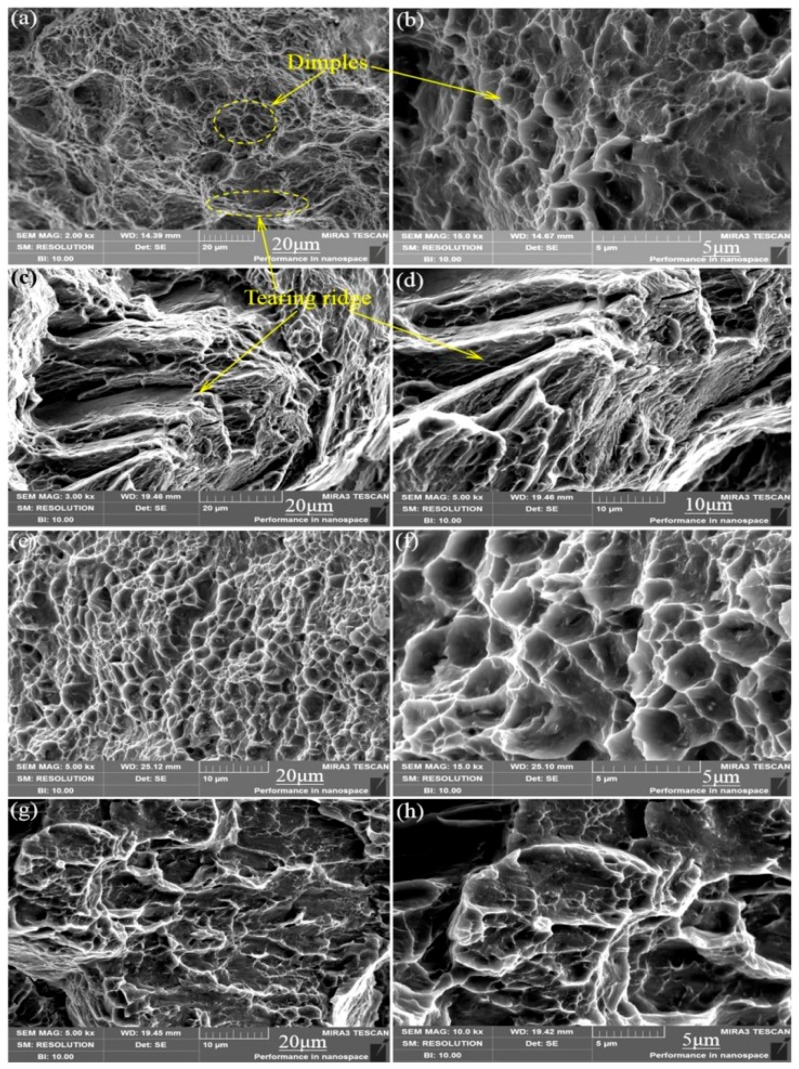
Microcosmic fracture morphology of as-cast and different-temperature extruded alloys: (**a**,**b**) As-cast; (**c**,**d**) 250 °C; (**e**,**f**) 300 °C; (**g**,**h**) 350 °C.

**Figure 9 materials-13-01147-f009:**
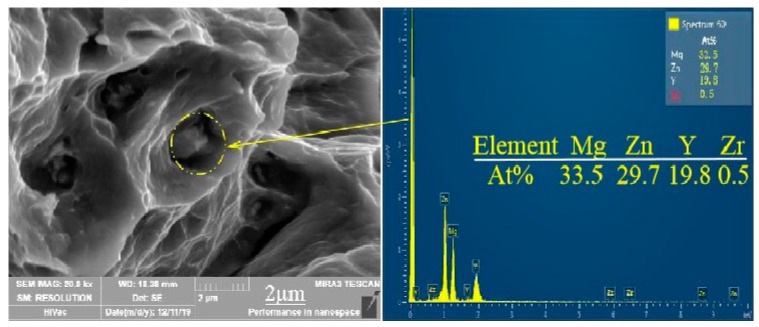
Microcosmic fracture morphology of extruded alloy at 300 °C and corresponding EDS results.

**Figure 10 materials-13-01147-f010:**
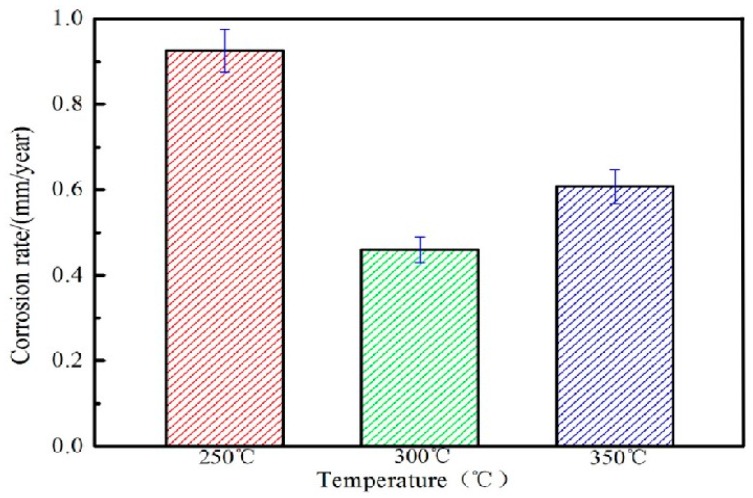
Loss-weight corrosion rate of alloy at different extrusion temperatures.

**Figure 11 materials-13-01147-f011:**
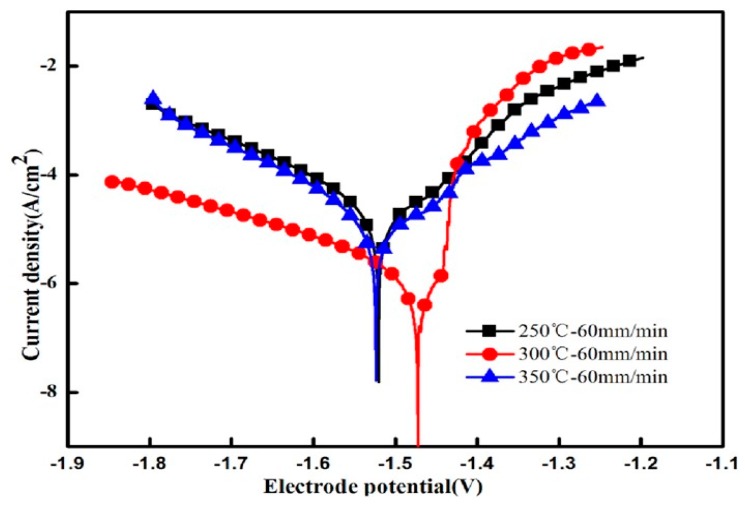
Electrochemical polarization curves of extruded alloys.

**Figure 12 materials-13-01147-f012:**
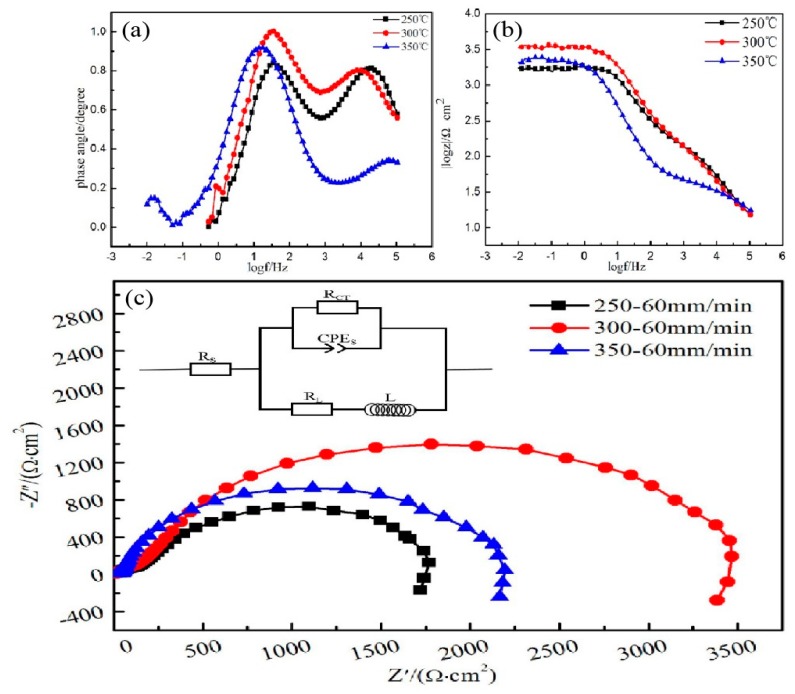
(**a**) Bode plots of phase angle versus frequency; (**b**) Bode plots of |z| vs. frequency, (**c**) Nyquist plots and equivalent circuits of the EIS spectra.

**Figure 13 materials-13-01147-f013:**
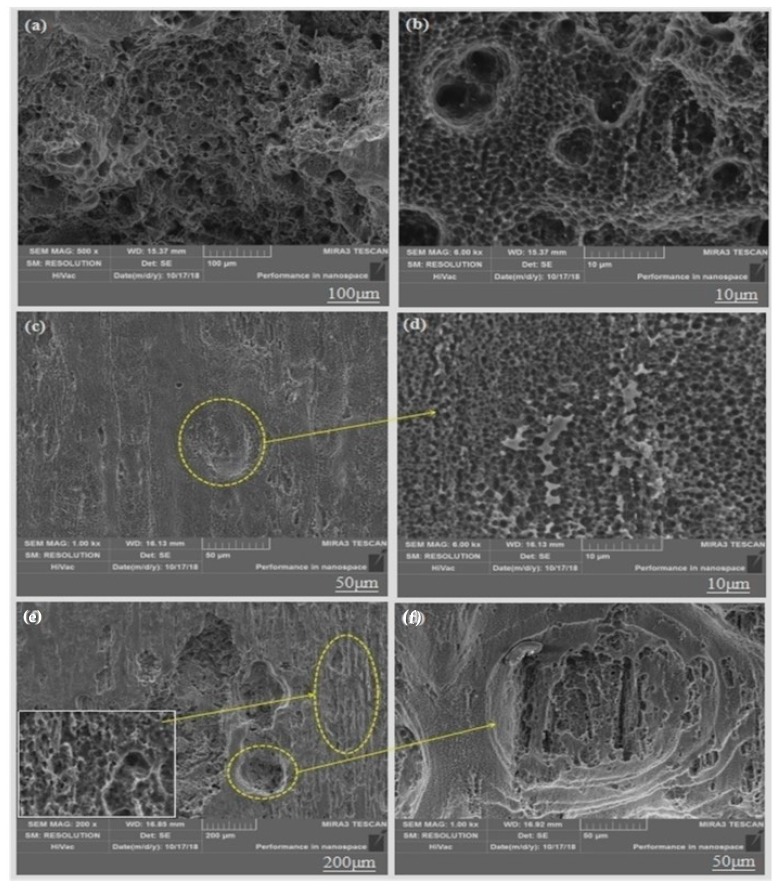
SEM image of surface morphology of alloys removed from corrosion products: (**a**,**b**) 250 °C; (**c**,**d**) 300 °C; (**e**,**f**) 350 °C.

**Figure 14 materials-13-01147-f014:**
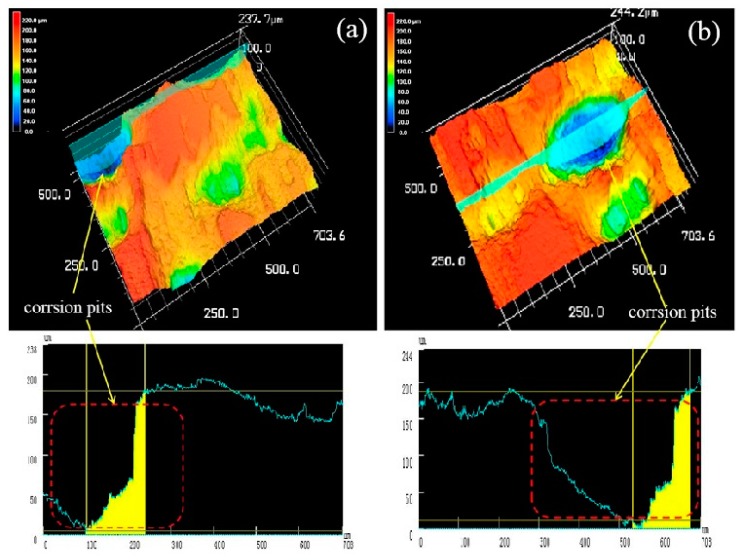
Both (**a**) and (**b**) were three-dimensional corrosion morphology of extruded alloy (250 °C) after corrosion product removal.

**Figure 15 materials-13-01147-f015:**
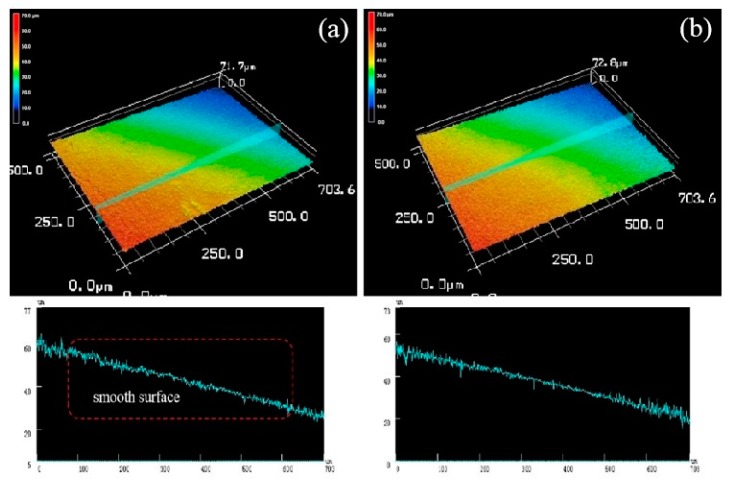
Both (**a**) and (**b**) were three-dimensional corrosion morphology of extruded alloy (300 °C) after corrosion product removal.

**Figure 16 materials-13-01147-f016:**
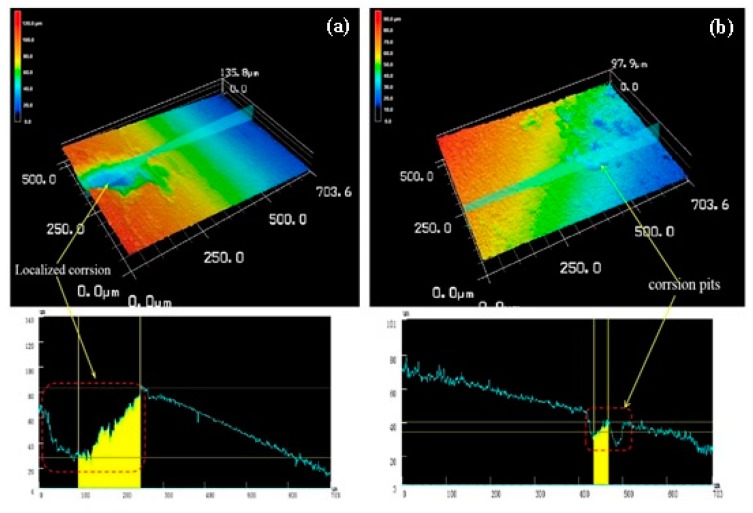
Both (**a**) and (**b**) were three-dimensional corrosion morphology of extruded alloy (350 °C) after corrosion product removal.

**Table 1 materials-13-01147-t001:** The EDS analysis of second phase in the as-cast and as-extruded alloys.

Point	Mg	Y	Zn	Zr	Phase
A	32.5	29.7	19.8	0.5	Mg_3_Zn_3_Y_2_
B	32.3	58.6	8.9	0.4	Mg_3_Zn_6_Y
C	45.7	18.2	36.5	0.1	Mg_3_Zn_3_Y_2_
D	31.6	10.9	57.2	0.3	Mg_3_Zn_6_Y
E	63.1	12.9	24.9	0	Mg_3_Zn_3_Y_2_
F	63.1	12.9	24.7	0	Mg_3_Zn_3_Y_2_
G	25.1	20.1	54.0	0.8	Mg_3_Zn_6_Y
H	59.9	12.6	27.3	0.2	Mg_3_Zn_3_Y_2_

**Table 2 materials-13-01147-t002:** Mechanical properties of alloys at different extrusion temperatures.

Temperature	Tensile Strength (MPa)	Yield Strength (MPa)	Elongation (%)
As-cast	238 ± 7	129 ± 6	13.8 ± 0.7
250 °C	372 ± 6	368 ± 5	10.8 ± 0.8
300 °C	321.6 ± 7	320.7 ± 4	25.4 ± 0.9
350 °C	272.5 ± 3	243 ± 3	22.5 ± 0.8

**Table 3 materials-13-01147-t003:** Related electrochemical parameters and weight loss corrosion rate.

Temperature	*I_corr_* (μA/cm^2^)	*E_corr_* (V/SCE)	*β_α_* (mV/decade)	*β_c_* (–mV/decade)	*R_P_* (kΩ cm^2^)	*P_i_* (mm/y)
250 °C	41.196	−1.5204	408.47	173.03	1.283	0.941
300 °C	16.29	−1.4725	267.47	248.5	3.438	0.372
350 °C	22.44	−1.5239	377.41	148.15	2.059	0.513

**Table 4 materials-13-01147-t004:** Fitting results obtained from the EIS spectra of extruded alloys.

Temperature	Rs (Ω·cm^2^)	Q (Ω^−1^S^n^/cm^2^)	N	R_ct_ (Ω·cm^2^)	L (H·cm^2^)
250 °C	66.72	1.523 × 10^−5^	0.82	846	632.7
300 °C	67.32	1.129 × 10^−5^	0.78	2198	1982.4
350 °C	64.15	1.432 × 10^−5^	0.84	1596	1105.3

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
