# Peer review of "Effects of Hot Extrusion Temperature on Mechanical and Corrosion Properties of Mg-Y-Zn-Zr Biological Magnesium Alloy Containing W Phase and I Phase"

_materials, 2020, doi:10.3390/ma13051147_

Round 1

Reviewer 1 Report

The paper is well written und addresses an interesting topic to be investigated, however, the authors have to improve the following issues:

-) please provide data on statistical analysis - this cannot be found

-) provide data on the amount of tests in each group.

-) please add a statistical section part in the manuscript

-) why was temperature chosen at 250°C, 300°C and 350°C ?

-) row 133: check the sentence (spelling)

-) row139: rephrase sentence

Author Response

X

Reviewer 2 Report

It is an interesting article on the mechanical and corrosion properties of extruded Mg-Zn-Y-Zr alloys. I recommend publishing the same after very minor structural changes.

The discussion on dynamic recrystallization is well written. For better clarity, I recommend EBSD analyses and DRX grain separation. Similarly, the deformation behavior and slip activation can be better understood from the EBSD texture analysis. Suggest presenting phase composition and distribution results before grain characteristics for a better flow of information. line 53: change 'smelting' into 'melting' Remove the sentence 'so does the corrosion resistance of the alloy' (line 106) while discussing microstructure as the corrosion results are only presented in later sections. Include references for Rx temperature of Mg-Zn-Y-Zr alloys and critical Y content for cytotoxicity Explain the purpose of testing flat samples and not cylindrical samples.

Author Response

G

Reviewer 3 Report

Abstract has to be rewritten.

Line 24 "Magnesium alloys have attracted more and more attention from many experts and scholars at home and abroad..." Please rephrase this.

Line 28 "Its modulus of elasticity (41-45Gpa) is the closest to the elastic modulus of 28 human bones [1-3], which grants it good biocompatibility." I have to disagree, this statement only does not grant a good biocompatibility. Please rephrase.

Introduction is messy. One can not understand clearly the aim of your study and there is no explanation for the "biological alloy" term used in the title. Which are the intended "biological" areas of use?

Please give details for manufacturers (city, country).

Please explain abbreviations when used first time.

Typing errors should be corrected (ex. Line 70, 72: ",Where:).

Line 83 "Some of the dense dotted second phase connecting and forming the strip at the grain boundary." Please rephrase, this is not a correct sentence.

Line 87 "In order to refine grains and improve tissue defects in the as-cast microstructure, and further increasing the density of the material." Same as previous.

The text has to be further checked for correct english. I will not further point out the language mistakes present, because there are too many.

Please place Figures (not Fig.) and Tables, together with captions or titles in the text, as requested in Instructions for authors. Please use same font size for all captions.

Figures 2b, f, h are not clear enough. Please provide better quality. Same for Figure 9 (scale not readable).

Do not number conclusions.

Please rewrite references according to Instructions for authors.

Author Response

X

Round 2

Reviewer 1 Report

Thank you for submitting the revised manuscript, however, statistical data is still missing. (number of tests, statistically significance?, which statistical tests did you use?...)

As previously mentioned, please add a statistical section part in the manuscript.

Author Response

Dear reviewer,

       I am sorry for my inaccurate impression leading to your confusion. The details have been uplaod according to your suggestion. Due to the cureent poor situation and my relative inferior personal comprehension, if my correction did not live up to your expection, please sent me suggestion again. 

                                                                                                   Thank you!
